# Squeezeformer: An Efficient Transformer for Automatic Speech Recognition

**Sehoon Kim**[*1], **Amir Gholami**[*1], **Albert Shaw**[†‡1], **Nicholas Lee**[†1], **Karttikeya Mangalam**[1], **Jitendra Malik**[1], **Michael W. Mahoney**[123], **Kurt Keutzer**[1]

[1]University of California, Berkeley  [2]ICSI  [3]LBNL
{sehoonkim, amirgh, nicholas_lee, mangalam, malik, mahoneymw, keutzer}@berkeley.edu
Albertshaw@google.com

## Abstract

The recently proposed Conformer model has become the *de facto* backbone model for various downstream speech tasks based on its hybrid attention-convolution architecture that captures both local and global features. However, through a series of systematic studies, we find that the Conformer architecture's design choices are not optimal. After re-examining the design choices for both the macro and micro-architecture of Conformer, we propose Squeezeformer which consistently outperforms the state-of-the-art ASR models under the same training schemes. In particular, for the macro-architecture, Squeezeformer incorporates (i) the Temporal U-Net structure which reduces the cost of the multi-head attention modules on long sequences, and (ii) a simpler block structure of multi-head attention or convolution modules followed up by feed-forward module instead of the Macaron structure proposed in Conformer. Furthermore, for the micro-architecture, Squeezeformer (i) simplifies the activations in the convolutional block, (ii) removes redundant Layer Normalization operations, and (iii) incorporates an efficient depthwise down-sampling layer to efficiently sub-sample the input signal. Squeezeformer achieves state-of-the-art results of 7.5%, 6.5%, and 6.0% word-error-rate (WER) on LibriSpeech test-other without external language models, which are 3.1%, 1.4%, and 0.6% better than Conformer-CTC with the same number of FLOPs. Our code is open-sourced and available online [25].

## 1 Introduction

The increasing success of end-to-end neural network models has been a huge driving force for the drastic advancements in various automatic speech recognition (ASR) tasks. While both convolutional neural networks (CNN) [19, 27, 29, 36, 61] and Transformers [24, 31, 32, 56, 57] have drawn attention as popular backbone architectures for ASR models, each of them has several limitations. Generally, CNN models lack the ability to capture global contexts and Transformers involve prohibitive computing and memory overhead. To overcome these shortcomings, Conformer [16] has recently proposed a novel convolution-augmented Transformer architecture. Due to its ability to synchronously capture global and local features from audio signals, Conformer has become the *de facto* model not only for ASR tasks, but also for various end-to-end speech processing tasks [17]. Furthermore, it has also achieved state-of-the-art performance in combination with recent developments in self-supervised learning methodologies as well [37, 62]. While the Conformer architecture was introduced as an autoregressive RNN-Transducer (RNN-T) [14] model in its original setting, it has been adopted with

---

[*]First author equal contribution.

[†]Second author equal contribution.

[‡]Now at Google.

36th Conference on Neural Information Processing Systems (NeurIPS 2022).

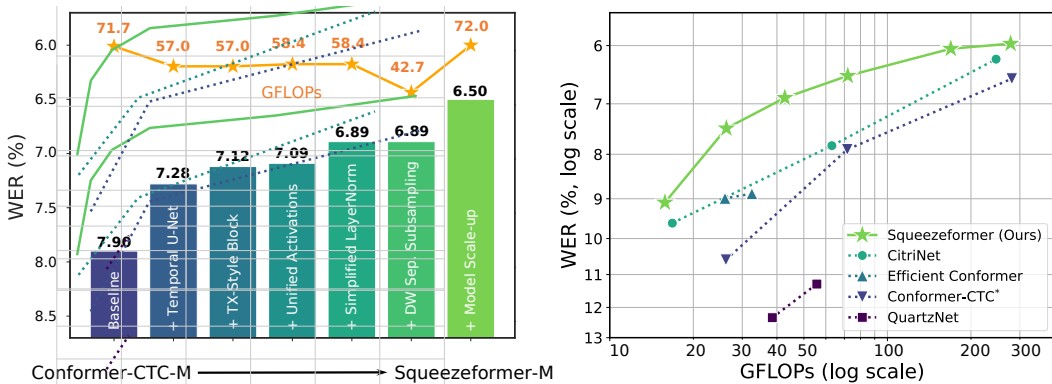

Figure 1: (Left) We perform a series of systematic studies on macro and micro architecture to redesign the Conformer architecture towards our Squeezeformer architecture. The bars and the line indicate the WER on LibriSpeech test-other dataset and the FLOPs, respectively. For each design modification, we strictly improve WER until our final Squeezeformer model outperforms Conformer by 1.40% WER improvement with the same number of FLOPs. See Tab. 1 for the details. (Right) LibriSpeech test-other WER vs. FLOPs for Squeezeformer and other state-of-the-art ASR models. Conformer-CTC* is our own reproduction to the best performance as possible and the others are the reported numbers in their papers [4, 27, 36]. Our architecture scales well to smaller and larger models to constantly outperform other models by a large margin throughout the entire FLOPs range. See Tab. 3 for the details. For both plots, the lower the WER, the better; however, we plotted in reverse for better visualization.

less critique to non-autoregressive schemes such as Connectionist Temporal Classification (CTC) [15] as well [38].

Despite being a key architecture in speech processing tasks, the Conformer architecture has some limitations that can be improved upon. First, Conformer still suffers from the quadratic complexity of the attention mechanism limiting its efficiency on long sequence lengths. This problem is further highlighted by the long sequence lengths of typical audio inputs as also pointed out in [46]. Furthermore, the Conformer architecture is relatively more complicated than Transformer architectures used in other domains such as in natural language processing [7, 44, 51] or computer vision [9, 10, 49]. For instance, the Conformer architecture incorporates multiple different normalization schemes and activation functions, the Macaron structure [34], as well as back-to-back multi-head attention (MHA) and convolution modules. This level of complexity makes it difficult to efficiently deploy the model on dedicated hardware platforms for inference [26, 39, 55]. More importantly, this raises the question of whether such design choices are necessary and optimal for achieving good performance in ASR tasks.

In this paper, we perform a careful and systematic analysis of each of the design choices with the goal of achieving lower word-error-rate (WER) for a given computational budget. We developed a much simpler and more efficient hybrid attention-convolution architecture in both its macro and micro-design that consistently outperforms the state-of-the-art ASR models. In particular, we make the following contributions in our proposed Squeezeformer model:

- We find a high temporal redundancy in the learned feature representations of neighboring speech frames especially deeper in the network, which results in unnecessary computational overhead. To address this, we incorporate the temporal U-Net structure in which a downsampling layer halves the sampling rate at the middle of the network, and a light upsampling layer recovers the temporal resolution at the end for training stability (§ 3.1.1).

- We redesign the hybrid attention-convolution architecture based on our observation that the back-to-back MHA and convolution modules with the Macaron structure are suboptimal. In particular, we propose a simpler block structure similar to the standard Transformer block [7, 51], where the MHA and convolution modules are each directly followed by a single feed forward module (§ 3.1.2).

- We finely examine the micro-architecture of the network and found several modifications that simplify the model overall and greatly improve the accuracy and efficiency. This includes (i) activation unification that replaces GLU activations with Swish (§ 3.2.1), (ii) Layer Normalization

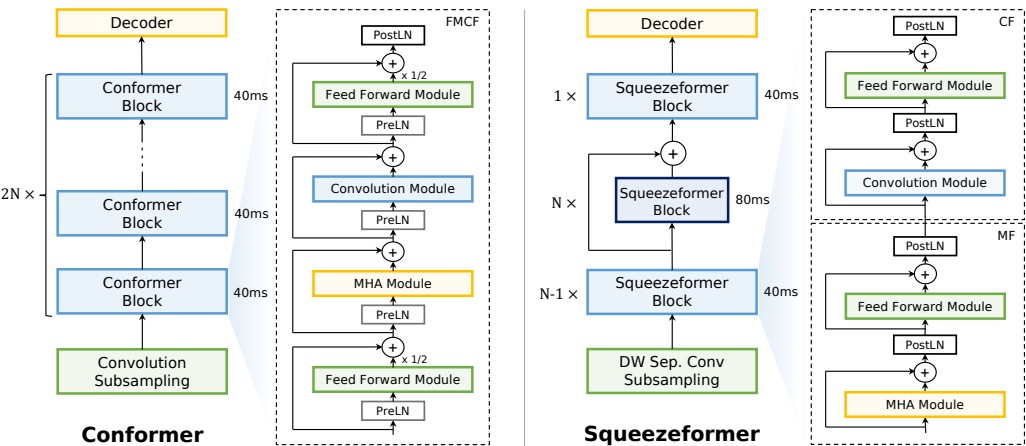

Figure 2: (Left) The Conformer architecutre and (Right) the Squeezeformer architecture which comprises of the Temporal U-Net structure for downsampling and upsampling of the sampling rate, the standard Transformer-style block structure that only uses Post-Layer Normalization, and the depthwise separable subsampling layer.

simplification by replacing redundant pre-Layer Normalization layers with a scaled post-Layer Normalization which incorporates a learnable scaling for the residual path that can be merged with other layers to be zero-cost during inference (§ 3.2.2), and (iii) incorporation of a depthwise separable convolution for the first sub-sampling layer that results in a significant floating point operations (FLOPs) reduction (§ 3.2.3).

- We show that the Squeezeformer architecture scales well with both smaller and larger models and consistently outperforms other state-of-the-art ASR models when trained under the same settings (Tab. 4.2, § 4.2). Furthermore, we justify the final model architecture of Squeezeformer with a reverse ablation study for the design choices (Tab. 4, § 4.3).

## 2 Related Work

The recent advancements in end-to-end ASR can be broadly categorized into (1) model architecture and (2) training methods.

**Model Architecture for End-to-end ASR.** The recent end-to-end ASR models are typically composed of an *encoder*, which takes as input a speech signal (i.e., sequence of speech frames) and extracts high-level acoustic features, and a *decoder*, which converts the extracted features from the encoder into a sequence of text. The model architecture of the encoder determines the representational power of an ASR model and its ability to extract acoustic features from input signals. Therefore, a strong architecture is critical for overall performance.

One of the popular choices for a backbone model architecture is convolutional neural network (CNN). End-to-end deep CNN models have been first explored in [29, 61], and further improved by introducing depth-wise separable convolution [21, 47, 50] in QuartzNet [27] and the Squeeze-and-Excitation module [23] in CitriNet [36] and ContextNet [19]. However, since CNNs often fail to capture global contexts, Transformer [51] models have also been widely adopted in backbone architectures due to their ability to capture long-range dependencies between speech frames [24, 31, 32, 56, 57]. Recently, [16] has proposed a novel model architecture named Conformer, which augments Transformers with convolutions to model both global and local dependencies efficiently. With the Conformer architecture as our starting point, we focus on designing a next-generation model architecture for ASR that is simpler, more accurate, and more efficient.

The hybrid attention-convolution architecture of Conformer has enabled the state-of-the-art results in many speech tasks. However, the quadratic complexity of the attention layer still proves to be cost prohibitive at larger sequence lengths. While different approaches have been proposed to reduce the cost of MHA in ASR [5, 46, 58, 59], their main focus is not changing the overall architecture design, and their optimizations can also be applied to our model, as they are orthogonal to the developments for Squeezeformer. Efficient-Conformer [4] introduces the progressive downsampling

Table 1: Starting from Conformer as the baseline, we redesign the architecture towards Squeezeformer through a series of systematic studies on macro and micro architecture. Note that for each design change, the WER on LibriSpeech test-clean and test-other datasets improves consistently. For comparison, we include the number of parameters and FLOPs for a 30s input in the last two columns.

| Model | Design change | test-clean | test-other | Params (M) | GFLOPs |
|---|---|---|---|---|---|
| Conformer-CTC-M | Baseline | 3.20 | 7.90 | 27.4 | 71.7 |
| | + Temporal U-Net (§ 3.1.1) | 2.97 | 7.28 | 27.5 | 57.0 |
| | + Transformer-style Block (§ 3.1.2) | 2.93 | 7.12 | 27.5 | 57.0 |
| | + Unified activations (§ 3.2.1) | 2.88 | 7.09 | 28.7 | 58.4 |
| | + Simplified LayerNorm (§ 3.2.2) | 2.85 | 6.89 | 28.7 | 58.4 |
| Squeezeformer-SM | + DW sep. subsampling (§ 3.2.3) | 2.79 | 6.89 | 28.2 | 42.7 |
| Squeezeformer-M | + Model scale-up (§ 3.2.3) | 2.56 | 6.50 | 55.6 | 72.0 |

scheme and grouped attention to reduce the training and inference costs of Conformer. Our work incorporates a similar progressive downsampling, but also introduces an up-sampling mechanism with skip connections from the earlier layers inspired by the U-Net [45] architecture in computer vision and the U-Time [43] architecture for sleep signal analysis. We find this to be critical for training stability and overall performance. In addition, through systematic experiments, we completely refactor the Conformer block by carefully redesigning both the macro and micro-architectures.

**Training Methodology for End-to-end ASR.** In the past few years, various self-supervised learning methodologies based on contrastive learning [2, 52, 62] or masked prediction [1, 6, 22] have been proposed to push forward the ASR performance. While a model pre-trained with self-supervised tasks generally outperforms when finetuned on a target ASR task, training strategies are not the main focus in this work as they can be applied independently to the underlying architecture.

# 3 Architecture Design

The Conformer architecture has been widely adopted by the speech community and is used as a backbone for different speech tasks. At a macro-level, Conformer incorporates the Macaron structure [34] comprised of four modules per block, as shown in Fig. 2 (Left). These blocks are stacked multiple times to construct the Conformer architecture. In this work, we carefully reexamine the design choices in Conformer, starting first with its macro-architecture, and then its micro-architecture design. We choose Conformer-CTC-M as the baseline model for the case study, and we compare word-error-rate (WER) on LibriSpeech test-other as a performance metric for each architecture. Furthermore, we measure FLOPs on a 30s audio input as a proxy for model efficiency. While we acknowledge that FLOPs may not always be a linear indicator of hardware and runtime efficiency, we choose FLOPs as it is hardware agnostic and is statically computable. However, we do measure the final end-to-end throughput of our changes, ensuring up to $1.34\times$ consistent improvement in runtime for different versions of Squeezeformer (Tab. 4.2)

## 3.1 Macro-Architecture Design

We first focus on designing the macro structure of Squeezeformer, i.e., how the blocks and modules are organized in a global scale.

### 3.1.1 Temporal U-Net Architecture

The hybrid attention-convolution structure enables Conformer to capture both global and local interactions. However, the attention operation has a quadratic FLOPs complexity with respect to the input sequence length. We propose to lighten this extra overhead by computing attention over a reduced sequence length. In the Conformer model itself, the input sampling rate is reduced from 10ms to 40ms with a convolutional subsampling block at the base of the network. However, this rate is kept constant throughout the network, with all the attention and convolution operations operating at a constant temporal scale.

To this end, we begin by studying the temporal redundancy in the learned feature representations. In particular, we analyze how the learned feature embeddings per speech frame are differentiated through

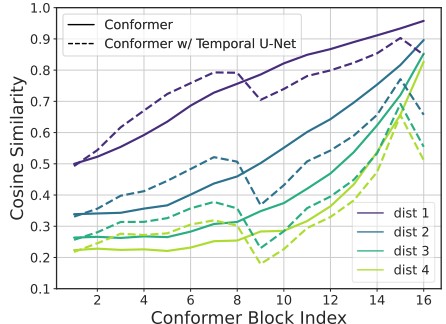

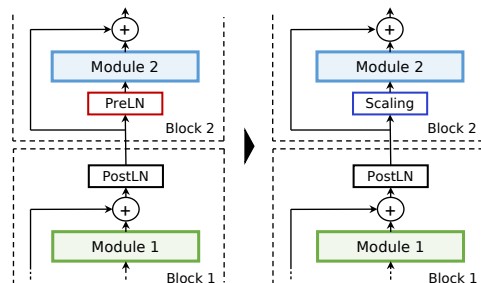

Figure 3: Cosine similarity between two embedding vectors of neighboring speech frames with varying adjacency distances across the Conformer blocks. The temporal dimension is downsampled after the 7th block and upsampled before the 16th block in the Temporal U-Net structure.

Figure 4: (Left) Back-to-back preLN and postLN at the boundary of the blocks. (Right) The preLN can be replaced with the learned scaling that readjusts the magnitude of the activation that goes into the subsequent module.

the Conformer model depth. We randomly sample 100 audio signals from LibriSpeech's dev-other dataset, and process them through the Conformer blocks, recording their per-block activations. We then measure the average cosine similarity between two neighboring embedding vectors. The results are plotted as the solid lines in Fig. 3. We observe that the embeddings for the speech frames directly next to each other have an average similarity of 95% at the topmost layer, and even those 4 speech frames away from each other have a similarity of more than 80%. This reveals that there is an increasing temporal redundancy as inputs are processed through the Conformer blocks deeper in the network. We hypothesize that this redundancy in feature embedding vectors causes unnecessary computational overhead and that the sequence length can be reduced deeper in the network without loss in accuracy.

As our first macro-architecture improvement step, we change the Conformer model to incorporate subsampling of the embedding vectors after it has been processed by the early blocks of the model. In particular, we keep the sample rate to be 40ms up to the 7th block, and afterwards we subsample to a rate of 80ms per input sequence by using a pooling layer. For the pooling layer we use a depthwise separable convolution with stride 2 and kernel size 3 to merge the redundancies across neighboring embeddings. This decreases the attention complexity by $4\times$ and also reduces the redundancies of the features. This temporal downsampling shares similarities with computer vision models, which often downsample the input image spatially to save compute and develop hierarchical level features [10, 20, 30, 48], and with the approach of Efficient Conformer [4].

However, the temporal downsampling alone leads to an unstable and diverging training behaviour (§ 4.3). One possible reason for this is the lack of enough resolution for the decoder after subsampling the rate to 80ms. The decoder maps an embedding for each speech frame into a single label, e.g., character, and therefore requires sufficient resolution for successful decoding of the full sequence. Inspired from successful architectures for dense prediction in computer vision such as U-Net [45], we incorporate the Temporal U-Net structure to recover the resolution at the end of the network through an upsampling layer as shown in Fig. 2. This upsampling block takes the embedding vectors processed by the 40ms and 80ms sampling rates, and produces an embedding with a rate of 40ms by adding them together via a skip connection. To the best of our knowledge, the closest work to our Temporal U-Net is the approach proposed in [43], in which the U-Net structure is incorporated into a fully-convolutional model to downsample sleep signals.

This change not only reduces the total FLOPs by 20% compared to Conformer[1], but also improves the test-other WER by 0.62% from 7.90% to 7.28% (Tab. 1, 2nd row). Furthermore, analyzing the cosine similarity shows that the Temporal U-Net architecture prevents the neighboring embeddings

---

[1]The total FLOPs is for the entire model. If we just study the attention block, the Temporal U-Net structure reduces the FLOPs by $2.31\times$ and $2.53\times$ FLOPs reduction for processing 30s and 60s audio signals as compared to Conformer-CTC-M baseline, respectively.

from becoming too similar to each others at the later blocks, in particular at the final block directly connected to the decoder, as shown in Fig. 3 as the dashed lines.

### 3.1.2 Transformer-Style Block

The Conformer block consists of a sequence of feed-forward ('F'), multi-head attention (MHA, 'M'), convolution ('C'), and another feed-forward module ('F'). We denote this as the FMCF structure. Note that the convolutional kernel sizes in ASR models are rather large, e.g., 31 in Conformer, which makes its behaviour similar to attention in mixing global information. This is stark contrast to convolutional kernels in computer vision, which often have small $3 \times 3$ kernels and hence benefit greatly from attention's global processing. As such, placing the convolution and MHA module with a similar functionality back-to-back (i.e., the MC substructure) does not seem prudent. Hence, we consider an MF/CF structure, which is motivated by considering the convolution module as a local MHA module. Furthermore, we drop the Macaron structure [34], as MHA modules followed by feed-forward modules have been more widely adopted in the literature [7, 9, 44, 51]. In a nutshell, we simplify the architecture to be similar to the standard Transformer network and denote the blocks MF and CF substructures, as shown in Fig. 2. This modification further improves the test-other WER by 0.16% from 7.28% to 7.12% and marginally improves the test-clean WER without affecting the FLOPs (Tab. 1, 3rd row).

## 3.2 Micro-Architecture Design

So far we have designed the macro structure of Squeezeformer by incorporating seminal architecture principles from computer vision and natural language processing into Conformer. In this subsection, we now focus on optimizing the micro structure of the individual modules. We show that we can further simplify the module architectures while improving both efficiency and performance.

### 3.2.1 Unified Activations

Conformer uses Swish activation for most of the blocks. However, it switches to a Gated Linear Unit (GLU) for its convolution module. Such a heterogeneous design seems over-complicated and unnecessary. From a practical standpoint, the use of multiple activations complicates hardware deployment, as an efficient implementation requires dedicated logic design, look up tables, or custom approximations [26, 39, 55]. For instance, on low-end edge devices with no dedicated vector processing unit, supporting additional non-linear operations would require additional look up tables or advanced algorithms [12, 13]. To address this, we propose to replace the GLU activation with Swish, unifying the choice of activation function throughout the entire model. We keep the expansion rate for the convolution modules. As shown in the 4th row of Tab. 1, this change does not entail noticeable changes in WER and FLOPs but only simplifies the architecture.

### 3.2.2 Simplified Layer Normalizations

Continuing our micro-architecture improvements, we note that the Conformer model incorporates redundant Layer Normalizations (LayerNorm), as shown in Fig. 4 (Left). This is because the Conformer model contains both a post-LayerNorm (postLN) that applies LayerNorm in between the residual blocks, as well as pre-LayerNorm (preLN) which applies LayerNorm inside the residual connection. While it is hypothesized that preLN stabilizes training and postLN benefits performance [53], these two modules used together lead to redundant back-to-back operations. Aside from the architectural redundancy, LayerNorm can be computationally expensive [26, 55] due to its global reduction operations.

However, we found that naïvely removing the preLN or postLN leads to training instability and convergence failure (§ 4.3). Investigating the cause of failure, we observe that a typical trained Conformer model has orders of magnitude differences in the norms of the learnable scale variables of the back-to-back preLN and postLN. In particular, we found that the preLN would scale down the input signal by a large value, giving more weight to the skip connection. Therefore, it is important to use a scaling layer when replacing the preLN component to allow the network to control this weight. This idea is also on par with several training stabilization strategies in other domains. For instance, NF-Net [3] proposed adaptive (i.e., learnable) scaling before and after the residual blocks to stabilize training without normalization. Furthermore, DeepNet [53] also recently proposed to add non-trainable rule-based scaling to the skip connections to stabilize preLN in Transformers.

Table 2: Detailed architecture configurations for Conformer-CTC (baseline) and Squeezeformer. For comparison, we include the number of parameters and FLOPs for a 30s input in the last two columns.

| Model | # Layers | Dimension | # Heads | Params (M) | GFLOPs |
|---|---|---|---|---|---|
| Conformer-CTC-S | 16 | 144 | 4 | 8.7 | 26.2 |
| Squeezeformer-XS | 16 | 144 | 4 | 9.0 | 15.8 |
| Squeezeformer-S | 18 | 196 | 4 | 18.6 | 26.3 |
| Conformer-CTC-M | 16 | 256 | 4 | 27.4 | 71.7 |
| Squeezeformer-SM | 16 | 256 | 4 | 28.2 | 42.7 |
| Squeezeformer-M | 20 | 324 | 4 | 55.6 | 72.0 |
| Conformer-CTC-L | 18 | 512 | 8 | 121.5 | 280.6 |
| Squeezeformer-ML | 18 | 512 | 8 | 125.1 | 169.2 |
| Squeezeformer-L | 22 | 640 | 8 | 236.3 | 277.9 |

Inspired by these computer vision advancements, we propose to replace preLN with a learnable scaling layer that scales and shifts the activations, $\text{Scaling}(x) = \gamma x + \beta$, with learnable scale and bias vectors $\gamma$ and $\beta$ of the size of feature dimension. For homogeneity of architectural design, we then replace the preLN throughout all the modules with the postLN-then-scaling as illustrated in Fig. 2 (Right) and make the entire model postLN-only. Note that the learned scaling parameters can be merged into the weights of the subsequent linear layer, as the architecture illustrated in Fig. 2 (Right), and hence have zero inference cost. With the learned scaling, our model further improves the test-other WER by 0.20% from 7.09% to 6.89% (Tab. 1, 5th row).

### 3.2.3 Depthwise Separable Subsampling

We now shift our focus from the Conformer blocks to the subsampling block. While it is easy to overlook this single module at the beginning of the architecture, we note that it accounts for a significant portion of the overall FLOPs count, up to 28% for Conformer-CTC-M with a 30-second input. This is because the subsampling layer uses two vanilla convolution operations each of which has a stride 2. To reduce the overhead of this layer, we replace the second convolution operation with a depthwise separable convolution while keeping the kernel size and stride the same. We leave the first convolution operation as is since it is equivalent to a depthwise convolution with the input dimension 1. This saves an additional 22% of the baseline FLOPs without a test-other WER drop and even a 0.06% improvement in test-clean WER (Tab. 1, 6th row). An important point to note here is that generally depthwise separable convolutions are hard to efficiently map to hardware accelerators, in part due its low arithmetic intensity. However, given the large FLOPs reduction, we consistently observe an overall improvement in the total inference throughput of up to 1.34× as reported in Tab. 3, as compared to the baseline Conformer models.

We name our final model with all these improvements as *Squeezeformer-SM*. Compared to Conformer-CTC-M, our initial baseline, Squeezeformer-SM improves WER by 1.01% from 7.90% to 6.89% with 40% less FLOPs. Given the smaller FLOPs of Squeezeformer-SM, we also scale up the model to a similar FLOPs cost as Conformer-CTC-M. In particular, we scale both depth and width of the model together following the practice in [8]. Scaling up the model achieves additional test-other WER gain of 0.39% from 6.89% to 6.50% (Tab. 1, 7th row), and we name this architecture *Squeezeformer-M*.

## 4    Results

### 4.1    Experiment Setup

**Models.** Following the procedure described in § 3, we construct Squeezeformer variants with different size and FLOPs: we apply the macro and micro-architecture changes in § 3.1 and § 3.2, respectively, to construct Squeezeformer-XS, SM, and ML from Conformer-S, M, and L, retaining the model size. Afterwards, we construct Squeezeformer-S, M, and L by scaling up each model to match the FLOPs of the corresponding Conformer. The detailed architecture configurations are described in Tab. 2.

While there are multiple options available for the decoder such as RNN-Transducer (RNN-T) [14] and Connectionist Temporal Classification (CTC) [15], we use a CTC decoder whose non-autoregressive

Table 3: WER (%) comparison on LibriSpeech dev and test datasets for Squeezeformer and other state-of-the-art CTC models for ASR including Conformer-CTC, QuartzNet [27], CitriNet [36], Transformer-CTC [31], and Efficient Conformer-CTC [4]. For comparison, we include the number of parameters, FLOPs, and throughput (Thp) on a single NVIDIA Tesla A100 GPU for a 30s input in the last three columns. *The performance numbers for Conformer-CTC are based on our own reproduction to the best performance as possible. All the other performance numbers are from the corresponding papers. †With and ‡without the grouped attention.

| Model | dev-clean | dev-other | test-clean | test-other | Params (M) | GFLOPs | Thp (ex/s) |
|---|---|---|---|---|---|---|---|
| Conformer-CTC-S* [16] | 4.21 | 10.54 | 4.06 | 10.58 | 8.7 | 26.2 | 613 |
| QuartzNet 5x5 [27] | 5.39 | 15.69 | - | - | 6.7 | 20.2 | - |
| Citrinet 256 [36] | - | - | 3.78 | 9.60 | 10.3 | 16.8 | - |
| Squeezeformer-XS | **3.63** | **9.30** | **3.74** | **9.09** | 9.0 | 15.8 | 763 |
| Conformer-CTC-M* [16] | 2.94 | 7.80 | 3.20 | 7.90 | 27.4 | 71.7 | 463 |
| QuartzNet 5x10 [27] | 4.14 | 12.33 | - | - | 12.8 | 38.5 | - |
| QuartzNet 5x15 [27] | 3.98 | 11.58 | 3.90 | 11.28 | 18.9 | 55.7 | - |
| Citrinet 512 [36] | - | - | 3.11 | 7.82 | 37.0 | 63.1 | - |
| Eff. Conformer-CTC† [4] | - | - | 3.57 | 8.99 | 13.2 | 26.0 | - |
| Eff. Conformer-CTC‡ [4] | - | - | 3.58 | 8.88 | 13.2 | 32.5 | - |
| Squeezeformer-S | 2.80 | 7.49 | 3.08 | 7.47 | 18.6 | 26.3 | 602 |
| Squeezeformer-SM | **2.71** | **6.98** | **2.79** | **6.89** | 28.2 | 42.7 | 558 |
| Conformer-CTC-L* [16] | 2.61 | 6.45 | 2.80 | 6.55 | 121.5 | 280.6 | 200 |
| Citrinet 1024 [36] | - | - | **2.52** | 6.22 | 143.1 | 246.3 | - |
| Squeezeformer-M | 2.43 | 6.51 | 2.56 | 6.50 | 55.6 | 72.0 | 431 |
| Squeezeformer-ML | **2.34** | **6.08** | 2.61 | **6.05** | 125.1 | 169.2 | 268 |
| Transformer-CTC [31] | 2.6 | 7.0 | 2.7 | 6.8 | 255.2 | 621.1 | - |
| Squeezeformer-L | **2.27** | **5.77** | **2.47** | **5.97** | 236.3 | 277.9 | 207 |

decoding method benefits training and inference latency [36]. However, the main focus of this work is the model architecture design of the encoder, which can be orthogonal to the decoder type.

Another subtlety when evaluating models is the use of external language models (LM). In many prior works [18, 24, 35, 40, 54, 56], decoders are often augmented with external LMs such as pre-trained 4-gram or Transformer, which boost the final WER by re-scoring the outputs in a more lexically accurate manner. However, we compare the results *without* external LMs to fairly compare the true representation power of the model architectures alone − external LMs can be incorporated as an orthogonal optimization afterward.

**Training Details.** Because the training recipes and codes for Conformer have not been open-sourced, we train it to reproduce the best performance numbers as possible. We train both Conformer-CTC and Squeezeformer on the LibriSpeech-960hr [41] for 500 epochs on Google's cloud TPUs v3 with batch size 1024 for the small and medium variants and 2048 for the large variants. We use AdamW [33] optimizer with weight decay 5e-4 for all models. More details for the training and evaluation setup are given in § A.1 and § A.2.

### 4.2 Main Results

In Tab. 3 we compare the WER of Squeezeformer with Conformer-CTC and other state-of-the-art CTC-based ASR models including QuartzNet [27], CitriNet [36], Transformer [31], and Efficient Conformer [4] on the clean and other datasets. Note that the performance numbers for Conformer-CTC[2] are based on our own reproduction to the best performance as possible due to the absence of public training recipes or codes. For simplicity, we denote WER as test-clean/test-other without % throughout the section.

**Squeezeformer vs. Conformer.** Our smallest model Squeezeformer-XS outperforms Conformer-CTC-S by 0.32/1.49 (3.74/9.09 vs. 4.06/10.58) with 1.66× FLOPs reduction. Compared with Conformer-CTC-M, Squeezeformer-S achieves 0.12/0.43 WER improvement (3.08/7.47 vs.

---

[2]The WER results exhibit some differences from the original paper [16] due to the difference in decoder. The original Conformer uses RNN-T decoder, which is known to generally result in better WER than CTC [4, 36, 60].

Table 4: Ablation studies for the design choices made in Squeezeformer, including Temporal U-Net, LayerNorm, and activation in the convolution module. *Without the upsampling layer, the model fails to converge.

| Ablation | Model | dev-clean | dev-other |
|---|---|---|---|
| Ours | Squeezeformer-M | 2.43 | 6.51 |
| Temporal U-Net (§ 3.1.1), | No skip connection | 2.78 | 7.38 |
| | No upsampling | N/A* | N/A* |
| LayerNorm (§ 3.2.2) | PostLN only | 5.60 | 14.00 |
| | PreLN only | 3.02 | 8.27 |
| Convolution module (§ 3.2.1) | No Swish | 2.53 | 6.73 |

3.20/7.90) with 1.47× smaller size and 2.73× less FLOPs, and Squeezeformer-SM further improves WER by 0.41/1.01 (2.79/6.89 vs. 3.20/7.90) with a comparable size and 1.70× less FLOPs. Compared with Conformer-CTC-L, Squeezeformer-M shows 0.24/0.05 WER improvement (2.56/6.50 vs. 2.80/6.55) with significant size and FLOPs reductions of 2.18× and 3.90×, respectively, and Squeezeformer-ML shows 0.19/0.50 WER improvement (2.61/6.05 vs. 2.80/6.55) with a similar size and 1.66× less FLOPs. Finally, our largest model Squeezeformer-L improves WER by 0.33/0.58 upon Conformer-CTC-L with the same FLOPs count, achieving the state-of-the-art result of 2.47/5.97.

**Squeezeformer vs. Other ASR Models.** As can be seen in Tab. 3, our model consistently outperforms QuartzNet, CitriNet, and Transformer with comparable or smaller model sizes and FLOPs counts. A notable result is a comparison against Efficient-Conformer: our model outperforms the efficiently-designed Efficient Conformer by a large margin of 0.79/1.99 (2.79/6.89 vs. 3.58/8.88) with the same FLOPs count. The overall results are summarized as a plot in Fig. 1 (Right) where Squeezeformer consistently outperforms other models across all FLOPs regimes.

## 4.3 Ablation Studies

In this section, we provide additional ablation studies for the design choices made for individual architecture components using Squeezeformer-M as the base model. See Tab. 4. Unless specified, we use the same hyperparameter settings as in the main experiment.

**Temporal U-Net.** In the 2nd row of Tab. 4, the model clearly underperforms by 0.35/0.87 without the skip connection from the downsampling layer to the upsampling layer. This shows that the high-resolution information collected in the early layers is critical for successful decoding. The 3rd row in Tab. 4 shows that our model completely fails to converge without the upsampling layer due to training stability, even with several different peak learning rates of {0.5, 1.0, 1.5}e-3.

**LayerNorm.** In the 4th line of Tab. 4, we show that WER drops significantly by 3.17/7.49 when we apply the PostLN-only scheme without the learned scaling layer. Another alternative design choice is to apply the PreLN-only scheme without the learned scaling, which also results in a noticeable WER degradation of 0.59/1.76 as shown in the 5th line of Tab. 4. In both cases, the model fails to converge, so we report the best WER before divergence. The results suggest that the learned scaling layer plays a key role for training stabilization and better WER.

**Convolution Module.** When ablating the GLU activation in the convolution modules, another possible design choice is to drop it without replacing it with the Swish activation. This, however, results in 0.10/0.22 worse WER as shown in the last line of Tab. 4.

## 5 Conclusions

In this work, we performed a series of systematic ablation studies on the macro and micro architecture of the Conformer architecture, and we proposed a novel hybrid attention-convolution architecture that is simpler and consistently achieves better performance than other models for a wide range of computational budgets. The key novel components of Squeezeformer's macro-architecture is the incorporation of the Temporal U-Net structure which downsamples audio signals in the second half of the network to reduce the temporal redundancy between adjacent features and save compute, as well as the MF/CF block structure similar to the standard Transformer-style which simplifies the architecture and improves performance. Furthermore, the micro-architecture of Squeezeformer simplifies the

activations throughout the model and replaces redundant LayerNorms with the scaled postLN, which is more efficient and leads to better accuracy. We also drastically reduce the subsampling cost at the beginning of the model by incorporating a depthwise separable convolution. We perform extensive testing of the proposed architecture and find that Squeezeformer scales very well across different model sizes and FLOPs regimes, surpassing prior model architectures when trained under the same settings. Our code along with the checkpoints for all of the trained models is open-sourced and available online [25].

## Acknowledgments

The authors acknowledge contributions from Dr. Zhewei Yao, Aniruddha Nrusimha, and Jiachen Lian. We also acknowledge gracious support from Google Cloud, Google TRC team, and specifically Jonathan Caton, Prof. David Patterson, and Dr. Ed Chi. We would also like to acknowledge the Sky Pilot team from UC Berkeley. Prof. Keutzer's lab is sponsored by Intel corporation, Intel VLAB team, Intel One-API center of excellence, as well as funding through BDD and BAIR. Sehoon Kim would like to acknowledge the support from Korea Foundation for Advanced Studies (KFAS). Amir Gholami was supported through funding from Samsung SAIT. Michael W. Mahoney would also like to acknowledge the UC Berkeley CLTC, ARO, NSF, and ONR. Our conclusions do not necessarily reflect the position or the policy of our sponsors, and no official endorsement should be inferred.

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
