# A   Appendix

## A.1   Training Details

For learning rate scheduling, we extend the widely used Noam Annealing [51] to additionally support the number of steps to maintain the peak learning rate $T_{\text{peak}}$ [46] and the decay rate $d$. That is, $\text{lr}(t) = \frac{\text{lr}_{\text{peak}}t}{T_0}$ for $t < T_0$, $\text{lr}_{\text{peak}}$ for $T_0 \le t < T_0 + T_{\text{peak}}$, and $\frac{\text{lr}_{\text{peak}}T_0^d}{(t-T_{\text{peak}})^d}$ for $t \ge T_0 + T_{\text{peak}}$, where $t$ is the step number, $\text{lr}_{\text{peak}}$ is the peak learning rate, and $T_0$ is the warmup steps. Note that the Noam annealing is a special case with $d = 0.5$ and $T_{\text{peak}} = 0$. We find warming up for 20 epochs, maintaining the peak learning rate for additional 160 epochs, and decaying with $d = 1$ work well in many cases, and fix these values throughout all experiments. We use the peak learning rate 2e-3, 1.5e-3, and {1, 0.5}e-3 for the small, medium, and large variants, respectively. We use SentencePiece [28] tokenizer with the vocabulary size 128, and the same dropout setting as in [16]. Finally, for data augmentation, we only use SpecAugment [42] with 2 frequency masks in [0, 27], and 5 (for all the small variants, Conformer-M and Squeezeformer-SM), 7 (for Squeezeformer-M) or 10 (for the large variants) time masks with the masking ratio of [0, 0.05].

## A.2   Evaluation Details

We evaluate the final models on both clean and other datasets using CTC greedy decoding. For both Conformer-CTC and Squeezeformer, we additionally measure the throughput on a single NVIDIA's Tesla A100 GPU machine (GCP a2-highgpu-1g instance) using 30s audio inputs as an indicator of hardware performance. Here, we use CUDA 11.5 and Tensorflow 2.5, and test with the largest possible batch size that saturates the machine.

## A.3   Transferrability to TIMIT

In Tab. A.1, we additionally evaluate the transferrability of Squeezeformer trained on LibriSpeech to unseen TIMIT [11] dataset with and without finetuning. In both cases, we used the same Sentence-Piece tokenizer as Librispeech training. For finetuning, we used the same learning rate scheduler as in § A.1 with the peak learning rate $\text{lr}_{\text{peak}}$ in {0.5, 1, 2, 5}e-4, 2 epochs of warmup ($T_0$), and 0 epoch of maintaining the peak learning rate ($T_{\text{peak}}$). All the other training recipes are the same as § A.1. We use Conformer-CTC as the baseline model to compare against, and we report WER measured on the test split. As can be seen in the table, the general trend aligns with the LibriSpeech results in Tab. 3: under smaller or same FLOPs and parameter counts, Squeezeformer outperforms Conformer-CTC, both with and without finetuning.

Table A.1: WER (%) comparison on TIMIT test split for Squeezeformer and Conformer-CTC that are trained on LibriSpeech with and without finetuning. For comparison, we also include the number of parameters and FLOPs.

| Model | without finetuning | with finetuning | Params (M) | GFLOPs |
|---|---|---|---|---|
| Conformer-S | 18.09 | 13.41 | 8.7 | 26.2 |
| Squeezeformer-XS | 16.31 | 12.89 | 9.0 | 15.8 |
| Conformer-M | 13.91 | 10.95 | 27.4 | 71.7 |
| Squeezeformer-S | 13.78 | 11.26 | 18.6 | 26.3 |
| Squeezeformer-SM | 13.65 | 10.50 | 28.2 | 42.7 |
| Conformer-L | 13.41 | 10.03 | 121.5 | 280.6 |
| Squeezeformer-M | 13.44 | 10.32 | 55.6 | 72.0 |
| Squeezeformer-ML | 11.35 | 9.96 | 125.1 | 169.2 |
| Squeezeformer-L | 12.92 | 9.76 | 236.3 | 277.9 |