# OpenReview forum: "Squeezeformer: An Efficient Transformer for Automatic Speech Recognition"
_NeurIPS.cc/2022/Conference — NeurIPS 2022 Accept_

### Official Review · Reviewer_TyAu · 2022-07-09

**Rating:** 8
**Confidence:** 5
**Soundness:** 4 excellent
**Presentation:** 4 excellent
**Contribution:** 3 good

**Summary:**

The paper considers the Conformer architecture popularly used in recent speech-input tasks. Under the CTC objective, they observe the following properties and motivate solutions to give Squeezeformer:
- Temporal redundancy of features motivates a U-Net downsample/upsample structure
- Adjacent PreLN/PostLNs motivate converting one to scale normalization
- Adjacent convolution/MHSAs motivate intervening FFNs

as well as other changes (unified activation fns., depthwise-separable convs.). They additively demonstrate each's improvements over Conformer-CTC and compare them under fixed GFLOPs or fixed parameter counts, and then at three overall model scales (vs. small, vs. medium, and vs. large models).

**Post-rebuttal: I increase my soundness and presentation scores from 3/4 to 4/4, and my score from 6/10 to 8/10.** This paper is an exemplar of what proposals of new deep-learning architectures should look like.

**Questions:**

- Are Squeezeformer's gains over Conformer possibly from optimizing for a different objective (from Transducer to CTC)? This limitation should be early and upfront.

- What if the reduction in cosine similarities is not meaningful, e.g., outputs give H__E_Y____ --> H_HEEY_Y_Y? What if U-Net outdoes downsampling for a simpler reason (the output token resolution argument?)

- Why do you think similarity increased (vs. baseline) before the downsampling?

- If unifying activations, why to Swish instead of GLU? (Also the point above re: multiple activations and efficiency)

Some comments re: presentation concerns above would also be appreciated.

**Limitations:**

The original Conformer was optimized for the bi-encoder Transducer objective and not for the uni-encoder CTC objective. For example, the Conformer architecture must also operate well on tokens (in the Transducer-specific label encoder), on which it is harder to e.g., justify downsampling. This work shows that Squeezeformer-CTC outdoes Conformer-CTC; it *does not show* Squeezeformer outdoes Conformer generally, not even in the Conformer's original setting.

(That said, the uni-encoder Conformer has become more typical, so these insights likely apply to many upcoming models.)

**Strengths And Weaknesses:**

Thanks to the authors for their extensive work.

While none of the individual methods are novel, their careful application atop Conformer-CTC is. The motivations are verbally clear, at least, and the CTC comparisons are fair (at fixed GFLOPs, fixed counts) as well as the ablations (e.g. vs postLN or preLN only). Re: significance, I'm convinced that Squeezeformer-CTC outperforms Conformer-CTC, and would expect this to extend to other _uni_-encoder speech applications given the margin of improvement (see Limitations). Re: clarity, the process of development and the individual methods are very clear.

However, some changes are motivated by debatable, possibly post-hoc "intuitions":
- In 3.1.1, cosine similarity in adjacent frames is touted as redundancy which is implied as bad, even in the last layer (L164-166). But redundant embeddings are good for the CTC objective, where consecutive equal predictions can express staying in the same token.
- Are multiple activation functions really a problem for efficient inference (L190)? There is still only one type of repeating block in the Conformer. Furthermore, Squeezeformer introduces new downsampling and upsampling layers, as well as variable memory consumption.

Also, some presentation concerns:
- I disagree with flipping the WER y-axis in Fig. 1. It is jarring as an ASR practitioner; also, it is reasonable to expect readers to intepret lower=better, visually (for some, the jarring reversed order of y-axis values may outweigh any gain in visual intuitiveness). Even worse, _the GFLOPs are superimposed atop this "higher-is-better" WER plot, but GFLOPs are a "lower-is-better" quantity!_
- In Table 3's last two rows, it is misleading to selectively include one non-CTC baseline. The original Conformer paper includes Transformer numbers (from your [22]) that are notably stronger (2.2, 5.6, 2.6, 5.7) than what you list. In fact, the original Conformer(-Transducer)'s numbers are far better (test = (2.1, 4.3)) with only 118.8M params.

Minor issues:
- FLOPs should defined before use, especially as it may be confused with FLOPS.
- A sentence or two about the scaling proposed in [9] in L241 would be helpful.
- L142-150 should mention that downsampling by itself is similar to Efficient Conformer
- L34: "it's" --> "its", "lengths" --> "lengths."
- L35: "inputs as alow pointed out"?
- L51: "doubles" --> "halves" (to match *down*sampling; the sampling rate is lower when fixed time is covered by fewer samples)
- L80, etc.: "depthwise", "depth-wise", "depthWise" all used
- L116, etc.: "Librispeech", "LibriSpeech" both used
- L219: "preLN-only" --> "postLN-only"
- Table 3: "an NVIDIA’s Tesla a100" --> "an NVIDIA Tesla A100"
- L422: "NVDIA" --> "NVIDIA"

---

> ### Author Response · Authors · 2022-08-02
> **Author Response to Reviewer TyAu (2/2)**
>
> > **Q4. (Presentation Concern 2)** In Table 3's last two rows, it is misleading to selectively include one non-CTC baseline. The original Conformer paper includes Transformer numbers (from your [22]) that are notably stronger (2.2, 5.6, 2.6, 5.7) than what you list. In fact, the original Conformer(-Transducer)'s numbers are far better (test = (2.1, 4.3)) with only 118.8M params.
>
> The Transformer that we used as our baseline is a CTC-based architecture (second to the last row in Table 3) as also mentioned in section 3.3.1 of [d]. We are also aware of the numbers reported in [e]; however, we did not compare those numbers with ours as [e] augmented external language model to decoding (section 4.3 and equation 15 in the paper) which in general results in a trivial WER improvement.
>
> > **Q5. (Questions 2)** What if the reduction in cosine similarities is not meaningful? What if U-Net outdoes downsampling for a simpler reason?
>
> Our observations and the results demonstrate that for successful decoding it is unnecessary to have a large amount of temporal redundancy as Conformer does, and by avoiding it we can achieve a better efficiency-accuracy trade-off. While it is not possible to give a definite answer due to the many moving parts involved in this problem, demystifying the impact of temporal redundancy on CTC decoding would be an interesting future research direction, as also mentioned in the answer to Q2.
>
> > **Q6. (Questions 3)** Why do you think similarity increased (vs. baseline) before the downsampling?
>
> This observation would be worth investigating further. For now, our intuitive explanation is that the downsampling layer introduces an explicit decoupling of the roles of the bottom and top layers such that the bottom layers focus more on high frequency features while the top ones focus on low frequency features. In such a case, we expect that the bottom layers embed time frames based on their local dependency, which makes their embeddings more similar to their neighboring frames. Another explanation could be that the bottom layers learn to compensate for the reduced redundancies after downsampling. Such downsampling operations are also known to produce different attention similarity patterns in vision tasks such as in MViT-v1 (Figure A.6) [f].
>
> > **Q7. (Questions 4)**  If unifying activations, why to Swish instead of GLU?
>
> GLU requires more complex memory transfer operations than Swish activation which is the reason we chose Swish instead. Note that Swish can be performed elementwise, while GLU requires applying a non-linear activation on the second half of the signal and multiplying it with the first half.
>
> \
> \
> References:
>
> [d] Likhomanenko et al. Rethinking Evaluation in ASR: Are Our Models Robust Enough? https://www.isca-speech.org/archive/pdfs/interspeech_2021/likhomanenko21_interspeech.pdf ([28] in our paper)
>
> [e] Karita et al. A Comparative Study on Transformer vs RNN in Speech Applications https://arxiv.org/pdf/1909.06317.pdf ([22] in our paper)
>
> [f] Fan et al. Multiscale Vision Transformers. https://arxiv.org/pdf/2104.11227.pdf

---

> > ### Comment · Reviewer_TyAu · 2022-08-07
> > **Reviewer TyAu Response to Author Response**
> >
> > Re: **Q4**, I understand now that you are only comparing CTC-style models in Tbale 3. it would then help to write "Transformer-CTC" / "Self Attention-CTC" and "Eff. Conformer-CTC" in its caption and model list, as well as "state-of-the-art CTC models for ASR". You can see how "WER (%) comparison on LibriSpeech ... for Squeezeformer ..., Transformer, and Efficient-Conformer" was misleading as the latter two can be read as the encoder-decoder Transformer and the Transducer-style Efficient Conformer, which as you now note in Footnote 2 are both known to be stronger than CTC, which is why the table made me skeptical.
> >
> > Re: **Q5** I can accept that the result is empirical.
> >
> > Good ideas raised re: **Q6**, and understood re: **Q7**.

---

> > > ### Author Response · Authors · 2022-08-09
> > > **Author Response to Reviewer TyAu (2/2)**
> > >
> > > >  **Q2-2.** I also mentioned it in case authors want to do a quick inspection of outputs (maybe Squeezeformer has fewer character deletions thanks to this? or now exhibits alternating behaviors like H_H_H_ that could be fixed -- many CTC-related algorithms rely on behaviors of the blank token). I do see Squeezeformer gaining wide adoption for encoder-only speech applications, so understanding such behaviors would help others down the line.
> > >
> > > This is an excellent question. We did some empirical analysis and found that the reduced cosine similarity is **not** because Squeezeformer is adding extra blank characters. In more detail, to better understand the impact of Squeezeformer to CTC decoding, we conducted several analyses on output characters that the CTC decoder produces. For this, we measured the number of
> > >
> > > 1. blank tokens (e.g., ‘_’),
> > > 2. repeating non–blank tokens (e.g., ‘aa’, ‘bb’),
> > > 3. transition from a non-blank token to a non-blank token (e.g., ‘aa’, ‘bb’, ‘ab’), and
> > > 4. transition from a non-blank token to a blank token (e.g., ‘a_’, ‘b_’)
> > >
> > > for the Conformer and Squeezeformer family on LibriSpeech test-clean and test-other datasets. The percentage of these numbers with respect to the total character counts on the entire dataset is presented in the tables below (nb: non-blank token, b: blank token).
> > >
> > > |   test-clean   | % b   | % repeating nb | % nb→nb | % nb→b |
> > > | ------------------- | ----- | -------------- | ------- | ------ |
> > > | Conformer-S  | 64.00 | 4.61   | 12.32   | 23.67  |
> > > | Conformer-M    | 63.82 | 4.77    | 12.41   | 23.76  |
> > > | Conformer-L   | 65.30 | 3.29   | 9.97    | 24.72  |
> > > | **Conformer (avg)**     | 64.37 | 4.22  | 11.56   | 24.05  |
> > > | Squeezeformer-XS    | 61.93 | 6.69 | 15.46   | 22.60  |
> > > | Squeezeformer-S     | 60.65 | 7.96 | 17.74   | 21.61  |
> > > | Squeezeformer-SM    | 60.61 | 7.99  | 17.63   | 21.75  |
> > > | Squeezeformer-M     | 60.58 | 8.01  | 17.98   | 21.44  |
> > > | Squeezeformer-ML    | 60.78 | 7.81   | 17.08   | 22.13  |
> > > | Squeezeformer-L     | 60.29 | 8.32   | 17.94   | 21.77  |
> > > | **Squeezeformer (avg)** | 60.80 | 7.79   | 17.30   | 21.88  |
> > >
> > >
> > > |    test-other   | % b   | % repeating nb | % nb→nb | % nb→b |
> > > | ------------------- | ----- | -------------- | ------- | ------ |
> > > | Conformer-S| 65.56 | 4.03| 11.74   | 22.69  |
> > > | Conformer-M| 65.29 | 4.25| 11.99   | 22.71  |
> > > | Conformer-L| 66.58 | 2.93| 9.77    | 23.65  |
> > > | **Conformer (avg)**     | 65.81 | 3.73| 11.16   | 23.01  |
> > > | Squeezeformer-XS    | 63.63 | 5.98| 14.70   | 21.66  |
> > > | Squeezeformer-S     | 62.42 | 7.10| 16.75   | 20.82  |
> > > | Squeezeformer-SM    | 62.35 | 7.15| 16.73   | 20.91  |
> > > | Squeezeformer-M     | 62.31 | 7.17| 17.06   | 20.63  |
> > > | Squeezeformer-ML    | 62.35 | 6.96| 16.11   | 21.35  |
> > > | Squeezeformer-L     | 62.08 | 7.41| 16.92   | 21.00  |
> > > | **Squeezeformer (avg)** | 62.52 | 6.96| 16.37   | 21.06  |
> > >
> > > For the test-clean dataset, Squeezeformer produced fewer blank tokens (3.5% less on average) compared to Conformer $\textit{across all}$ models in the same model family. In addition, Squeezeformer shows more transitions from a non-blank token to another non-blank token (5.74% more on average) as well as a larger number of repeating tokens (3.57% higher on average). On the contrary, we see a decrease in the number of non-blank to blank token transitions (2.17% less on average). We observed an identical trend with the test-other dataset.
> > >
> > > In summary, Squeezeformer does not tend to produce extra blank tokens as compared to Conformer, and thus the reason for the increased cosine similarity is not because of producing extra blank tokens.
> > >
> > > > **Q3.** understood. As I (and presumably many speech folks reading this paper) are less familiar with hardware details, you could replace L192 with a more self-contained example, such as: "...multiple activations complicates hardware deployment; e.g., on low-end edge devices with no dedicated vector processing units, supporting additional non-linear operations requires additional lookup tables or advanced algorithms"
> > >
> > > This is a fair point. We added **L195** to further elaborate on this point.
> > >
> > > > **Q4.** I understand now that you are only comparing CTC-style models in Tbale 3. it would then help to write "Transformer-CTC" / "Self Attention-CTC" and "Eff. Conformer-CTC" in its caption and model list, as well as "state-of-the-art CTC models for ASR". You can see how "WER (%) comparison on LibriSpeech ... for Squeezeformer ..., Transformer, and Efficient-Conformer" was misleading as the latter two can be read as the encoder-decoder Transformer and the Transducer-style Efficient Conformer, which as you now note in Footnote 2 are both known to be stronger than CTC, which is why the table made me skeptical.
> > >
> > > We clarified these in the caption and the entries of Table 3.
> > >
> > >
> > > Finally, we appreciate the reviewer’s comment on the presentation concern regarding Figure 1. We are investigating the best practice for visualization.

---

> > > > ### Comment · Reviewer_TyAu · 2022-08-09
> > > > **Reviewer TyAu Response to Additional Author Response**
> > > >
> > > > Thanks for the final round of edits; I believe the presentation (once Figure 1 is also iterated on) is now fair and very clear. I also appreciate the authors' extensive insights and experiments: here, showing the improved blank/non-blank token repetitions of Squeezeformer-CTC over Conformer-CTC, and to the other reviewers (continuing their paper's trend of precise ablations).
> > > >
> > > > **I increase my soundness and presentation scores from 3/4 to 4/4, and my score from 6/10 to 8/10.** This paper is an exemplar of what proposals of new deep-learning architectures should look like.

---

> > > ### Author Response · Authors · 2022-08-09
> > > **Author Response to Reviewer TyAu (1/2)**
> > >
> > > We appreciate the reviewer’s feedback that helps us improve the paper. We acknowledge the reviewer’s concerns and clarified them in our revised version of the paper (marked blue).
> > >
> > > > **Q1.** I agree it is reasonable that you did not run Transducer experiments. I would like to see this early, as a phrase in the abstract and/or an extra sentence in L41 like "Furthermore, the Conformer architecture was optimized for the ASR-specific Transducer model, but has been adopted with less critique to encoder-only schemes like CTC or self-supervised pretraining."
> > >
> > > We brought this point upfront in **L31** to help readers notice that our main focus is on the encoder-only Conformer with CTC decoder.
> > >
> > > >  **Q2-1.** perhaps a sentence in L167 acknowledging the paradox of reduced temporal dependency at the final layer still improving CTC would help.
> > >
> > > Although it is not added in the revised paper due to the page limit, we will add the following sentence in **L169** of the final version of the paper:
> > >
> > > While a certain amount of redundancy might help cluster subsequent time frames into the same label, our observation demonstrates that the large amount of time redundancy as in Conformer is unnecessary for successful decoding, and by avoiding this, a better efficiency-accuracy tradeoff can be achieved.

---

> ### Author Response · Authors · 2022-08-02
> **Author Response to Reviewer TyAu (1/2)**
>
> We appreciate the reviewer's valuable comments. Responses to your questions are provided below.
>
> > **Q1. (Limitation 1 / Question 1)** Lack of experimental justification with RNN-T Decoder: The original Conformer was optimized for the bi-encoder Transducer objective and not for the uni-encoder CTC objective. For example, the Conformer architecture must also operate well on tokens (in the Transducer-specific label encoder), on which it is harder to e.g., justify downsampling. This work shows that Squeezeformer-CTC outdoes Conformer-CTC; it does not show Squeezeformer outdoes Conformer generally, not even in the Conformer's original setting.
>
> We did not perform experiments with RNN-T due to the limited compute resources and, therefore, our claims in the paper have been strictly about CTC models which are widely used in production today. We do not claim superior performance with RNN-T decoder, neither in the manuscript nor in rebuttal. We will clarify this in the final version of the paper. Having said this, CTC ASR models are indeed getting increased attention in industry due to the need to process audio signals in the data centers for applications such as offline ASR.
>
> > **Q2. (Weakness 1)** In 3.1.1, cosine similarity in adjacent frames is touted as redundancy which is implied as bad, even in the last layer (L164-166). But redundant embeddings are good for the CTC objective, where consecutive equal predictions can express staying in the same token.
>
> A certain amount of redundancy might be needed for clustering subsequent time frames into the same label and therefore improving the CTC decoding performance. At the same time, it might have a negative impact from the standpoint of efficiency and end-to-end latency. Reducing the FLOPs associated with the U-Net subsampling also allows one to use deeper or wider models which in turn may result in higher accuracy. Therefore, it is important to find the right tradeoff. Our results show the noticeable accuracy (WER) improvement under the same (or even smaller) computational cost through downsampling, which demonstrates the positive impact of increased efficiency from downsampling is greater than the potential negative impact that reduced redundancy and temporal information would have caused. Under a more drastic downsampling scheme than the proposed method, this trend may have been flipped and the accuracy (WER) could have been degraded due to the lack of enough redundancy and temporal information as like the reviewer’s claim. Investigating the impact of temporal redundancy on successful CTC decoding and searching for the optimal amount of redundancy would be an interesting future research direction.
>
> > **Q3. (Weakness 2)** Are multiple activation functions really a problem for efficient inference (L190)? There is still only one type of repeating block in the Conformer. Furthermore, Squeezeformer introduces new downsampling and upsampling layers, as well as variable memory consumption.
>
> The overhead of supporting non-linear operations is actually pretty important for deployment on low-end edge devices which often do not contain dedicated vector processing units found in server-grade GPUs. One popular practice has been a lookup table that stores pre-computed outputs of non-linear functions where supporting multiple non-linear operations would require multiple lookup tables or advanced algorithms. Because non-linear operations in DL applications (as in Conformer/Squeezeformer’s case) are often on the critical path, supporting fast lookup tables often entails hardware cost (e.g., area and complexity), and reducing this hardware cost is an active research area [a, b, c].
>
> However, new downsampling/upsampling layers such as those in Squeezeformer do not require extra logic or dedicated look-up tables in hardware. While it involves variable memory consumption that might require more involved hardware/mapping optimizations, its drastic reduction in number of FLOPs (2x) will eventually benefit the inference cost as can be seen in the end-to-end latency improvement of Table 3.
>
> \
> \
> References:
>
> [a] Geng et al. Hardware-aware Exponential Approximation for Deep Neural Networks. https://openreview.net/pdf?id=Sksl1mJPM
>
> [b] Geng et al. Hardware-aware Softmax Approximation for Deep Neural Networks. https://oar.a-star.edu.sg/storage/p/pv0k3qeq26/0421.pdf
>
> [c] Yu et al. NN-LUT: Neural Approximation of Non-Linear Operations for Efficient Transformer Inference. https://arxiv.org/pdf/2112.02191.pdf

---

> > ### Comment · Reviewer_TyAu · 2022-08-07
> > **Reviewer TyAu Response to Author Response**
> >
> > Re: **Q1**, I agree it is reasonable that you did not run Transducer experiments. I would like to see this early, as a phrase in the abstract and/or an extra sentence in L41 like "Furthermore, the Conformer architecture was optimized for the ASR-specific Transducer model, but has been adopted with less critique to encoder-only schemes like CTC or self-supervised pretraining."
> >
> > Re: **Q2**, perhaps a sentence in L167 acknowledging the paradox of reduced temporal dependency at the final layer still improving CTC would help. I also mentioned it in case authors want to do a quick inspection of outputs (maybe Squeezeformer has fewer character deletions thanks to this? or now exhibits alternating behaviors like H_H_H_ that could be fixed -- many CTC-related algorithms rely on behaviors of the blank token). I do see Squeezeformer gaining wide adoption for encoder-only speech applications, so understanding such behaviors would help others down the line.
> >
> > Re: **Q3**, understood. As I (and presumably many speech folks reading this paper) are less familiar with hardware details, you could replace L192 with a more self-contained example, such as: "...multiple activations complicates hardware deployment; e.g., on low-end edge devices with no dedicated vector processing units, supporting additional non-linear operations requires additional lookup tables or advanced algorithms"
> >
> > Finally, I reiterate my concern re: "flipping the WER y-axis" in Figs. 1 and 2, and how it flips directionality of WER and GFLOPs in Fig. 1. (Please give your readers more credit!)

---

### Official Review · Reviewer_xM6i · 2022-07-11

**Rating:** 6
**Confidence:** 5
**Soundness:** 2 fair
**Presentation:** 4 excellent
**Contribution:** 3 good

**Summary:**

The paper improves upon a Conformer model for speech recognition. The paper finds experimentally that the subsequent activations are are highly correlated in the higher confomer blocks. Therefore, it proposes to subsample in the the temporal dimension. Then, the paper proposes to remove the Macaron block. Finally it proposes to optimize the micro-architecture of the conformer: to use the Swish activation instead of GLU, use the PostLN with an extra scaling, and use a separable convolution for the first layer.

The paper describes in detail the motivation for each improvement. Then, the experiments show that adding each change one-by-one gradually improves the performance of the proposed model.

Finally, the paper reports the performance of the proposed model in comparison to prior publications.

**Questions:**

Suggestions
- Please fix the typos
- Add the previously reported conformer result into the table
- Evaluating on another dataset would strengthen the paper
- It seems that the suggestion to remove the Macaron architecture contradicts the original study https://arxiv.org/pdf/2005.08100.pdf Table 3. Am I missing something or this paper proposes to undo this change in the original?

**Strengths And Weaknesses:**

# Strengths

The paper is very well written and is a delight to read. I was able to follow the motivations for each step of the improvement and the proposed change.

That's being said, each proposal is sound and makes sense.  The experimentation is rigorous and follows the best practices.

# Weaknesses

My main criticism is that the baseline has much worse performance than it is reported in [0]. As far as I can see the main difference between the 6.8% (test-other) reported here and 4.3% in [0] is the decoder. While I understand, that the baseline source code is not available, the paper needs to be more upfront about the differences and better performance of [0]. Perhaps, it can be included in the table as "closed source SOTA with RNN-T decoder" or something like this.

There is a paper which predates Jasper as an end-to-end CNN architecture: [1].

[0] https://arxiv.org/pdf/2005.08100.pdf
[1] https://arxiv.org/pdf/1701.02720.pdf

# Typos

page 7, 219 p -> alpha
Page 8, line 255 work -> works
page 8, line 258 "-" -> "--"

---

> ### Author Response · Authors · 2022-08-02
> **Author Response to Reviewer xM6i (2/2)**
>
>
> > **Q2. (Question 3)** Evaluating on another dataset would strengthen the paper
>
> We thank the reviewer for the suggestion. To address this concern, we have conducted an additional set of experiments on transferring Squeezeformer trained on Librispeech to TIMIT with and without finetuning. In both cases, we used the same sentence piece tokenizers as Librispeech training. For finetuning, we used the same learning rate scheduling with peak learning rate ($lr_{peak}$) in {0.5, 1, 2, 5}e-4, 2 epochs of warmup ($T_0$) and 0 epoch for maintaining the peak learning rate ($T_{peak}$). The WER is measured on the test split. The results are as follows:
>
> |                  | without finetuning | with finetuning | Params (M) | FLOPs (G) |
> | ---------------- | ---------------- | ------------- | ---------- | --------- |
> | Conformer-S      | 18.09            | 13.41         | 8.7        | 26.2      |
> | Squeezeformer-XS | 16.31            | 12.89         | 9.0        | 15.8      |
> | Conformer-M      | 13.91            | 10.95         | 27.4       | 71.7      |
> | Squeezeformer-S  | 13.78            | 11.26         | 18.6       | 26.3      |
> | Squeezeformer-SM | 13.65            | 10.50         | 28.2       | 42.7      |
> | Conformer-L      | 13.41            | 10.03         | 121.5      | 280.6     |
> | Squeezeformer-M  | 13.44            | 10.32         | 55.6       | 72.0      |
> | Squeezeformer-ML | 11.35            | 9.96          | 125.1      | 169.2     |
> | Squeezeformer-L  | 12.92            | 9.76          | 236.3      | 277.9     |
>
> As can be seen in the table, the general trend is similar to the Librispeech result in Table 3: under smaller or same FLOPs and parameter counts, Squeezeformer outperforms Conformer. This empirically shows the transferability of Squeezeformer to an unseen/different ASR dataset. We included this Table in the supplementary of the revised paper (please check Section A.4 and Table A.2).
>
> > **Q3. (Question 4)**  It seems that the suggestion to remove the Macaron architecture contradicts the original study https://arxiv.org/pdf/2005.08100.pdf Table 3. Am I missing something or this paper proposes to undo this change in the original?
>
> In the Conformer paper, the detailed model architectures for ablations are not clearly described. However, given that Table 3 of the Conformer paper shows the impact of removing individual components from Conformer towards the vanilla Transformer, we can assume that the 4th row in the table is comparing the performance of (1) Transformer with relative positional embedding vs. (2) Transformer with relative positional embedding and Macaron structure, both of which don’t contain convolution blocks (i.e., MF structure vs. FMF structure). This is different from how we disentangle the Macaron structure from the Conformer structure (i.e., FMCF structure vs. FCMF structure), and is not contradictory to our intuition that having an FFN layer after MHA and convolution layers benefit performance. Further, in our experience, the architectural components often have non-linear interactions such that the overall ablation path is non-conservative in the final performance. Ablating a component, say, B after ablating A might show no drop in performance while ablating B directly on the original architecture can cause significant degradation and vice-versa.

---

> ### Author Response · Authors · 2022-08-02
> **Author Response to Reviewer xM6i (1/2)**
>
> We appreciate the reviewer's valuable comments. Responses to your questions are provided below.
>
> > **Q1. (Weakness 1 / Question 2)** My main criticism is that the baseline has much worse performance than it is reported in the original Conformer paper. As far as I can see the main difference between the 6.8% (test-other) reported here and 4.3% in the original paper is the decoder. (Add the previously reported conformer result into the table)
>
>
> We acknowledge the reviewer’s point that there is a performance gap between the reported numbers and our reproduction for the Conformer baseline. The difference mainly arises from the difference in decoder: please note that we are comparing Conformer-CTC and not RNN-T decoder which is used in the original Conformer paper. RNN-T decoders are generally known to result in better (lower) WER results, which have been widely observed and studied in prior literature [a, b, c]. Furthermore, there is the known difficulty of reproducibility of Conformer results as well. We’ve indeed made a considerable effort in reproducing Conformers despairing any looming bugs that might affect our own results as well. Unfortunately, the authors have not open-sourced their implementation, and the difficulty of reproducing their results has been a known problem in many prior works due to the absence of publicly available training codes and recipes. For this reason, prior works have also reported and compared against their own Conformer results trained under fair training conditions [c, d, e]. We mention upfront the performance gap against the original Conformer in Section 4.2 of the revised paper.
>
> Under these considerations, we believe the more reasonable baseline for us is the Conformer-CTC checkpoints from NVIDIA’s open-source library Nemo [f], whose training codes and recipes are public as well. Below is a new table with the Nemo results for Conformer-S, M, and L appended. We note that, compared to our own Conformer implementation, Nemo implementation has two major differences that enhance performance: (1) Nemo’s Conformer S and M configurations are larger than ours (18 layers, 176 hidden dim for S and 18 layers, 256 hidden dim for M) which result in a larger number of FLOPs as can be seen in the table; (2) Nemo has been trained for 1000 epochs whereas ours are trained for 500 epochs.
> |Model| test-clean (%)| test-other (%)|FLOPs (G)| #training epochs |
> | ------------------ | ---- | ---- | ----- | ----- |
> | Conformer-S (Nemo) | 3.4  | 8.8  | 39.6  | 1000  |
> | Squeezeformer-S    | 3.08 | 7.47 | 26.3  | 500   |
> | Conformer-M        | 3.20 | 7.90 | 71.7  | 500   |
> | Conformer-M (Nemo) | 3.0  | 7.3  | 78.2  | 1000  |
> | Squeezeformer-SM   | 2.79 | 6.89 | 42.7  | 500   |
> | Squeezeformer-M    | 2.56 | 6.50 | 72.0  | 500   |
> | Conformer-L        | 2.80 | 6.55 | 280.6 | 500   |
> | Conformer-L (Nemo) | 2.7  | 6.1  | 280.6 | 1000  |
> | Squeezeformer-ML   | 2.61 | 6.05 | 169.2 | 500   |
> | Squeezeformer-L    | 2.47 | 5.97 | 277.9 | 500   |
> | Squeezeformer-L*   | 2.44 | 5.65 | 277.9 | 1000* |
>
>
>
> Compared to the Nemo’s results, the general trend of Squeezeformer achieving lower WER than Conformer in similar FLOPs regime remains the same. Moreover, under the fair condition with the same number of training epochs (i.e., 1000), Squeezeformer shows even further improvement as can be seen in the last column in the table (marked *), which further emphasizes the strength of Squeezeformer over Conformer. Due to the page limit, we will include this new comparison to Table 3 in the final version of the paper.
>
> \
> \
> References:
>
> [a] Zhang et al. Benchmarking LF-MMI, CTC and RNN-T Criteria for Streaming ASR. https://arxiv.org/abs/2011.04785
>
> [b] Majumdar et al. Citrinet: Closing the Gap between Non-Autoregressive and Autoregressive End-to-End Models for Automatic Speech Recognition https://arxiv.org/pdf/2104.01721.pdf
>
> [c] Berchi et al. Efficient Conformer: Progressive Downsampling and Grouped Attention for Automatic Speech Recognition. https://arxiv.org/pdf/2109.01163.pdf
>
> [d] Shim et al. Understanding the Role of Self Attention for Efficient Speech Recognition. https://openreview.net/pdf?id=AvcfxqRy4Y
>
> [e] Lohrenz et al. Relaxed Attention: A Simple Method to Boost Performance of End-to-End Automatic Speech Recognition, https://arxiv.org/pdf/2107.01275.pdf
>
> [f] NVIDIA Nemo. https://github.com/NVIDIA/NeMo

---

### Official Review · Reviewer_6bur · 2022-07-26

**Rating:** 6
**Confidence:** 5
**Soundness:** 4 excellent
**Presentation:** 3 good
**Contribution:** 3 good

**Summary:**

This paper proposes Squeezeformer, a novel hybrid attention-convolution architecture for ASR through a series of extensive architectural studies and improvements over the Conformer architecture. Note, Conformer has been the de-facto architecture for E2E speech processing tasks and Squeezeformer simplifies the shows impressive improvements on the Conformer-CTC architecture. The authors also have open sourced the code and trained model checkpoints which should be supremely helpful to the community in re-using and building on these models.

**Questions:**

* I enjoyed reading through the paper, I wonder if the authors studied the positional embeddings in Conformer and thought of any improvements in a positional encoding scheme for self-attention in the Squeezeformer?

**Strengths And Weaknesses:**

Strengths:
 *  The authors extensively study the architectural design of Conformer and simplifies a lot of the choices in Conformer building out a simpler, cleaner and possibly better architecture, Squeezeformer.
* The paper introduces U-Net style temporal scaling architecture and are able to show strong results at various different model scales improving upon earlier Conformer benchmarks which have been SOTA for possibly all speech processing tasks.
* Given that Conformer has been the de-facto architecture for speech over the last few years, the results jumps in performance by Squeezeformer are significant and the extensive set of ablations and experiments in this paper are very helpful in providing insights to the community.

Weaknesses:
* Even though the results are impressive, there is not much originality in this paper as it builds upon extensively studying the existing Conformer architecture and combines standard temporal downsampling tricks from vision. But still, the results shown are strong and the experiments are significant that would be of huge help to the community.

---

> ### Author Response · Authors · 2022-08-02
> **Author Response to Reviewer 6bur**
>
> We appreciate the reviewer's valuable comments. Responses to your questions are provided below.
>
> > **Q1. (Weakness 1)** Even though the results are impressive, there is not much originality in this paper as it builds upon extensively studying the existing Conformer architecture and combines standard temporal downsampling tricks from vision.
>
> While the individual methods might not be regarded as novel, it is the first work to extend and carefully adjust these techniques to the ASR domain, which combined together to result in a significant performance improvement (both in WER and computational efficiency) over the de-facto Conformer architecture, as well as improved latency.
>
> > **Q2. (Question 1)**  I wonder if the authors studied the positional embeddings in Conformer and thought of any improvements in a positional encoding scheme for self-attention in the Squeezeformer?
>
> While relative positional embeddings in Conformer involve additional computation costs as compared to absolute positional embeddings, we empirically observed that replacing them with absolute positional embeddings resulted in noticeable performance degradation. Developing better positional embedding schemes has been an active research area (not just in ASR, but across domains) including designing new schemes [a], or new self-attention mechanisms which do not require positional embeddings at all [b]. Our scope for analyzing the positional embedding was limited to absolute/relative embeddings. However, we do expect that Squeezeformer would benefit the same as other models with new positional embeddings in further research.
>
> \
> \
> References:
>
> [a] Likhomanenko et al. CAPE: Encoding Relative Positions with Continuous Augmented Positional Embeddings. https://ronan.collobert.com/pub/2021_cape_arxiv.pdf
>
> [b] Shim et al. Similarity and Content-based Phonetic Self Attention for Speech Recognition. https://arxiv.org/pdf/2203.10252.pdf

---

### Author Response · Authors · 2022-08-02
**Author Response to the Reviewers**

We appreciate all the reviewers for taking the time to review our work, and providing us with their valuable feedback.
We provided responses to the questions that each of the reviewers has commented on, and uploaded the revised version of the paper and supplementary material with the typos and minor/major fixes addressed.

---

### Meta-Review · Area_Chair_e1wz · 2022-08-26

**Recommendation:** Accept
**Confidence:** Certain

**Metareview:**

The paper conducts thorough analysis of the Conformer architecture and brings insights and techniques from other fields to simplify and improve the model structure, which is also demonstrated to show nice gains. Though as pointed by reviewers the novelty is limited, the study is very useful to the field.

**Award:**

No

---

### Decision · Program_Chairs · 2022-09-14

Accept